# Depressive Symptoms, Fatigue and Social Relationships Influenced Physical Activity in Frail Older Community-Dwellers during the Spanish Lockdown due to the COVID-19 Pandemic

**DOI:** 10.3390/ijerph18020808

**Published:** 2021-01-19

**Authors:** Laura M. Pérez, Carmina Castellano-Tejedor, Matteo Cesari, Luis Soto-Bagaria, Joan Ars, Fabricio Zambom-Ferraresi, Sonia Baró, Francisco Díaz-Gallego, Jordi Vilaró, María B. Enfedaque, Paula Espí-Valbé, Marco Inzitari

**Affiliations:** 1Parc Sanitari Pere Virgili, Area of Intermediate Care, 08023 Barcelona, Spain; ccastellano@perevirgili.cat (C.C.-T.); lsoto@perevirgili.cat (L.S.-B.); jars@perevirgili.cat (J.A.); minzitari@perevirgili.cat (M.I.); 2RE-FiT Barcelona Research Group, Vall d’Hebron Institute of Research & Parc Sanitari Pere Virgili, 08023 Barcelona, Spain; sbaro@perevirgili.cat (S.B.); pespi@perevirgili.cat (P.E.-V.); 3GIES Research Group, Basic Psychology Department, Autonomous University of Barcelona, 08193 Bellaterra, Spain; 4Geriatric Unit, IRCCS Istituti Clinici Scientifici Maugeri, 20138 Milano, Italy; matteo.cesari@unimi.it; 5Department of Clinical Sciences and Community Health, Università di Milano, 20138 Milano, Italy; 6Navarrabiomed, Complejo Hospitalario de Navarra (CHN), Universidad Pública de Navarra (UPNA), IdiSNA, 31008 Pamplona, Navarra, Spain; fabricio.zambom.ferraresi@navarra.es; 7Primary Healthcare Center Larrard, Atenció Primària Parc Sanitari Pere Virgili, 08023 Barcelona, Spain; 8Primary Healthcare Center Bordeta-Magòria, Institut Català de la Salut, 08014 Barcelona, Spain; fdiazg.bcn.ics@gencat.cat; 9Department of Health Sciences, Blanquerna—Ramon Llull University, 08022 Barcelona, Spain; jordivc@blanquerna.url.edu; 10Institut Català de la Salut, Gerència de Barcelona, 08007 Barcelona, Spain; menfedaque.bcn.ics@gencat.cat; 11Department of Medicine, Autonomous University of Barcelona, 08035 Barcelona, Spain

**Keywords:** COVID-19, frailty, aging, physical activity, mental health, social relationships

## Abstract

Due to the dramatic impact of the COVID-19 pandemic, Spain underwent a strict lockdown (March–May 2020). How the lockdown modified older adults’ physical activity (PA) has been poorly described. This research assesses the effect of the lockdown on PA levels and identifies predictors of sufficient/insufficient PA in frail older community-dwellers. Community-dwelling participants from the +ÀGIL Barcelona frailty intervention program, suspended during the pandemic, underwent a phone-assessment during the lockdown. PA was measured before and after the lockdown using the Brief Physical Activity Assessment Tool (BPAAT). We included 98 frail older adults free of COVID-19 (mean age = 82.7 years, 66.3% women, mean Short Physical Performance Battery = 8.1 points). About one third of participants (32.2%) were not meeting sufficient PA levels at the end of the lockdown. Depressive symptoms (OR = 0.12, CI95% = 0.02–0.55) and fatigue (OR = 0.11, CI95% = 0.03–0.44) decreased the odds of maintaining sufficient PA, whereas maintaining social networks (OR = 5.07, CI95% = 1.60–16.08) and reading (OR = 6.29, CI95% = 1.66–23.90) increased it. Living alone was associated with the reduction of PA levels (b = −1.30, CI95% = −2.14–−0.46). In our sample, pre-lockdown mental health, frailty-related symptoms and social relationships were consistently associated with both PA levels during-lockdown and pre-post change. These data suggest considering specific plans to maintain PA levels in frail older community-dwellers.

## 1. Introduction

The COVID-19 global pandemic has had a dramatic impact on the population’s health, especially for older adults [1]. To mitigate the quick spread of infection, several measures have been undertaken around the globe. In Spain, one of the most affected countries, these measures included a strict lockdown (14 March to 2 May 2020). During this period, citizens were not allowed to leave their homes except to attend work, essential medical appointments, shop for food and take care of vulnerable or dependent individuals. A steady phase of de-escalation was then implemented until 21 June 2020, when mobility restrictions were finally removed.

Frailty, a dynamic state of increased vulnerability to internal or external stressors, determines a higher risk of negative health outcomes, such as disability, falls, fractures, institutionalization and death [2]. Therefore, its identification and the development of individualized prevention strategies are mandatory [3,4,5]. In order to promote a more comprehensive and life-course assessment of older adults, the World Health Organization (WHO) introduced the concept of functional ability (i.e., having the capabilities that enable all people to be and do what they value), which is determined by the interaction between intrinsic capacity (i.e., composite of all physical and mental capacities) and the environment [3]. This latter was clearly altered by the COVID-19 pandemic, and the consequent preventative restrictions. Although lockdowns and mobility restrictions are crucial public health countermeasures, these caused a radical and sudden change in people’s lifestyles, in particular regarding physical activity (PA) levels [6,7].

PA has been previously described as a risk factor for frailty [8,9,10] and a key component of interventions to prevent or reduce the development and progression of frailty [3,11,12,13]. It has been estimated that the preventive measures applied during the COVID-19 pandemic led to a 25% reduction of PA in the general population [6,14,15,16,17], and more than 45% in older adults [6,14,15,16,18,19]. Despite these data, the possible determinants of this reduction in PA levels have not been explored yet. This might be relevant to design future strategies to resume PA and prevent frailty and disability. Comprehensive geriatric assessment (CGA) includes measures pertaining to different domains, such as functional, physical, cognitive, mood, nutritional and social, which usually interact to determine negative health outcomes for older adults [20]. Variables of the CGA might help predict the change or decrease in PA levels in older adults during the lockdown.

Among the heterogeneous group of older adults, the COVID-19 pandemic posed particular challenges to community-dwelling frail older adults’ approach and care. Nevertheless, the impact and consequences of decreased daily activities and social contacts limitations in this vulnerable group, including community-dwelling, frail older adults with a relatively preserved autonomy before the pandemic, has been poorly described. Focusing on this population group is particularly relevant due to the increased risk of accelerated disability. Therefore, it is crucial to appropriately target at-risk individuals to implement individualized post-pandemic plans to recover PA.

In this paper, we describe PA changes due to mobility restrictions in community-dwelling, frail older persons who had not been diagnosed with COVID-19 from a running program, to delay or revert frailty in community-dwelling older adults of Barcelona. Taking advantage of the extensive CGA pre-lockdown, which also included a standardized measure of PA, we explored factors associated with the improvement or maintenance of sufficient PA levels during the lockdown.

## 2. Materials and Methods

### 2.1. Study Population

The study population was derived from the +ÀGIL Barcelona project, an implemented, ongoing, real-life multidimensional intervention program, based on integrating primary care, geriatrics and other community resources. Models and results of the initiative have been previously published [11,21]. In brief, the program enrolls nondisabled frail older adults [22] based on the comprehensive geriatric assessment (CGA) performed by a geriatric multidisciplinary team in collaboration with primary care professionals for designing a person-tailored community intervention. Pillars of the intervention include a 10-week boost of multicomponent physical exercise, aiming to empower participants to perform PA, complemented with home sessions based on the validated ViviFrail platform [23]. After the boost, the continuation of PA in existing resources in the community is pursued. Promotion of the Mediterranean diet, health education and optimization of pharmacological therapies are also part of the intervention. After the initial CGA, the geriatrician repeats an assessment at three months (and occasionally six months) to revise and adapt the intervention. +ÀGIL Barcelona has been continuously running from July 2016 until March 2020 (enrolling 100 participants/year). Due to the COVID-19 pandemic, face-to-face assessments were temporarily suspended, replaced by phone calls during the follow-up procedure and data collection.

In May 2020, at the end of the Spanish lockdown applied by the Spanish Government (14 March to 2 May), a follow-up visit via phone was performed with each participant in the +ÀGIL Barcelona program who had been assessed face-to-face during the 12 months prior to the lockdown (either as the baseline, three or six-month visit). In case the participant could not complete the phone call assessment, a self-identified proxy or caregiver answered the follow-up interview. The interviews lasted around 20 min and were performed by two trained physiotherapist researchers.

### 2.2. Measure of Physical Activity

During the phone survey, the level of PA was assessed with the Brief Physical Activity Assessment Tool (BPAAT), the same tool used in all the routine visits pre-lockdown [24,25]. The BPAAT is a two-question tool. The first item explores the frequency and duration of PA at vigorous intensity, and the second item assesses the frequency and PA duration at moderate-intensity during a typical week. The BPPAT scoring algorithm was designed to identify whether patients meet or not PA recommendations through the combination of both questions. Its total score ranges from 0 to 8, allowing the ability to distinguish sufficiently active (20 min of vigorous-intensity ≥ 3 times/week or 30 min of moderate-intensity ≥ 5 times/week or ≥5 times/week of any combination of moderate or vigorous PA, scores 4–8 points) from insufficiently active participants (who do not meet any previous recommendation, scores 0–3 points). Previous studies report a reliability of 0.76 and construct validity of 0.71 [25]. The outcomes of interest were: (1) total PA during the lockdown (BPAAT total score at the phone survey); (2) improvement (from insufficient to sufficient) or maintenance of sufficient PA vs reduction (from sufficient or insufficient) or maintenance of insufficient PA, according to BPAAT total score. Qualitative aspects related to PA during the lockdown were also part of the phone interview (e.g., self-reported maintenance of pre-lockdown PA level, use of +ÀGIL Barcelona strategies to maintain physical activity).

### 2.3. Covariates

Data from the last face-to-face CGA pre-lockdown were considered as covariates. These included sociodemographic data (age, sex, education, living alone), clinical characteristics including the Charlson Comorbidity Index [26] and current treatment, functional independence for basic (ADLs) and instrumental activities for daily living (IADLs), nutrition, depression, physical function and frailty. Functional independence for ADLs was assessed by the Barthel index, an ordinal scale range from 0–100 points (total dependent-independent) [27]. The Lawton index was used to measure the independence for IADLs; it ranges from 0–8 points (total dependent-independent) [28]. Nutrition was assessed by the Mini Nutritional Assessment–Short Form, a validated screening tool to identify older adults who are malnourished or at risk of malnutrition; it ranges from 0–12 points (normal nutrition status: 14–12 points, at risk malnutrition: 11–8 points, malnourish: 0–7 points) [29]. The Mini-cog© (Washington, DC, USA), a 3-min instrument was used for cognitive impairment screening, range from 0–5 points (<3 points increase the likelihood of dementia or cognitive impairment) [30,31]. The screening of depression symptoms was assessed by the Yesavage Geriatric Depression Scale, a simple and valid tool for discriminating depressive symptoms; it ranges from 0–15 points (≥6 points: moderate depression) [32,33]. The physical function was measured with the Short Physical Performance Battery (SPPB), a tool that combines the results of the gait speed, chair stand and balance tests, with a range from 0–12 points (<10 points high likelihood of frailty) [34]. Finally, the frailty degree was assessed according to the Clinical Frailty Scale (CFS), a clinical judgement-based frailty tool, which summarizes the CGA results and generates a frailty score range from very-fit to terminally ill [35]. The validity and reliability of all the scales used have been assessed previously.

Data collected by semi-structured phone interview during the lockdown, included sociodemographic data (cohabitation, support at home, social relations with family or other persons, tools to maintain social contact and frequency), COVID-19 related variables (COVID-19 diagnosis on relatives, new onset of acute clinical events and self-reported fatigue, considered a frailty-related symptom [36], health visits canceled due to the pandemic, communication with healthcare professional, and activities to stay active during the lockdown.

### 2.4. Statistical Analysis

Characteristics of the sample before the lockdown are presented as mean values and standard deviation (SD), or median values and interquartile range (IQR) for continuous variables, as applicable, and frequency and percentages for categorical variables. The pre-post lockdown PA level was analyzed by a paired sample *t*-test for repeated samples when total BPAAT score was taken into account, and McNemar’s test for a repeated sample when the change in PA categories was analyzed (sufficient vs insufficient PA level). Differences among participants with improvement or sufficient PA level and those with reduction or insufficient PA level, were analyzed using the Student’s *t*-test or the Mann-Whitney U-test and Chi-square test, as appropriate. Variables showing an association with the outcomes (*p*-value < 0.05) and those considered clinically relevant, or to have a potential influence on the outcomes, were included in a stepwise multivariable logistic (dichotomous outcome of change) and stepwise linear regression models (total PA during the lockdown), as appropriate, to obtain final parsimonious models (with age, gender and education locked into the models for being relevant predictors of PA or proxy of socio-economic status). All analyses were performed using Stata version 14.

### 2.5. Ethical Aspects

The +ÀGIL Barcelona program and study protocol were approved by the Clinical Research Ethics Committee of the Institut Universitari d’Investigació en Atención Primaria, Jordi Gol i Gorina (20/048-P). Before starting the telephone interview, oral informed consent was obtained from all participants or, if the participant could not provide such consent, from a proxy.

## 3. Results

Out of 117 contacted participants from +ÀGIL, 107 (91.5%) agreed to answer the phone survey. To ensure the homogeneity of the population, those previously diagnosed with SARS-COVID-19 (*n* = 4), or with incomplete PA data (*n* = 5), were excluded. Finally, we included in the analyses 98 participants (mean age = 82.4 SD 6.1 years; 66.3% women; mean time since last face-to-face visit 8.1 SD 3.7 months). The vast majority (88.8%) of the phone interviews were answered by the participants. There were no significant differences in terms of age, sex and time since the last face-to-face assessment between those who participated and those who refused to participate or were excluded from the survey.

A general decrease in PA level during the lockdown (BPAAT total score: −1.1/8 (95 CI% 0.6; 1.5) points; *p* < 0.001)) and reduction of participants reporting sufficient PA (−32.2%; *p* = 0.003) was reported (Figure 1). Overall, 22% of the sample continued to follow the personalized PA recommendations designed and delivered through the +ÀGIL Barcelona program.

Participants with reduced or insufficient PA presented higher pre-lockdown IADLs disability and comorbidity, more prevalent depressive symptoms and previous diagnosis or positive screening for cognitive impairment/dementia than those who improved or maintained sufficient PA (Table 1). This same group, with reduced or insufficient PA level, also reported more fatigue, more health concerns and less social contact with friends or other people outside the family during the lockdown (Table 2). On the other hand, participants who improved or maintained sufficient PA were more likely to follow PA-related recommendations from the +ÀGIL Barcelona program during the lockdown and to perform other leisure activities, such as reading, as a strategy to stay physically or mentally active.

In multivariable models, living alone before the lockdown (ß= −1.30, 95%CI −2.14–−0.46, *p* = 0.003), previous depressive symptoms (ß= −1.15, 95%CI −1.89–−0.41, *p* = 0.003) and self-reported fatigue during the COVID-19 outbreak (ß= −1.25, 95%CI −1.87–−0.63, *p* < 0.001) were inversely associated with PA levels (BPAAT total score) during the lockdown. Having social contact with people different from family (ß = 0.99, 95%CI 0.41–1.57, *p* = 0.001) and performing reading activities during the lockdown (ß = 0.74, 95%CI 0.08–1.39, *p* = 0.028) were associated with higher BPAAT scores during the lockdown (Table 3). Neither physical function measures (SPPB, gait speed), nor frailty (CFS) or cognitive impairment were associated with the amount of PA during the lockdown or with the change in PA levels.

Looking at the pre-post change in PA levels, multivariable models showed consistent results for pre-lockdown depressive symptoms (OR= 0.12, 95%CI 0.02–0.55, *p* = 0.006) and self-reported fatigue (OR = 0.11, 95%CI 0.03–0.44; *p* = 0.002), which were negatively associated with improvement/maintenance of sufficient PA, as well as for social contacts with people different from family networks (friends or neighbors), which increased the odds for a positive outcome (OR = 5.07, 95%CI 1.60–16.08; *p* = 0.006). In this model, reading during the lockdown (OR = 6.29, 95%CI 1.66–23.90; *p* = 0.007) was positively associated with improving/maintaining sufficient PA (Table 3).

## 4. Discussion

In our population of community-dwelling frail older adults, strict home lockdown due to the COVID-19 pandemic determined a generalized decrease in PA, although a remarkable proportion maintained or improved PA. Regarding pre-lockdown characteristics, higher depressive symptoms were associated with total PA during the outbreak and change in PA, and participants living alone performed less PA during the outbreak. On the other hand, social relationships and leisure activities during the outbreak were directly associated with PA levels and pre-post change, whereas self-reported fatigue had an inverse association with PA levels.

During the first months of the outbreak, Spain adopted a strict home lockdown motivated by the pandemic’s severe impact [37]. Previous studies from Italy and Japan also reported a decrease in PA [6,14,15,16,17,18,19,38]; or a rising prevalence of inactive older persons [18,19]. Despite the similarities in the mobility restriction measures among the three countries, the study populations are different: the Italian study used a cohort that underwent the implantation of a cardio meter-defibrillator before the pandemic [14,18] whereas the one enrolled by Suzuki et al. [18] was discharged from a rehabilitation setting; both samples were significantly younger than ours. Compared with the study by Yamada et al., which showed a relevant prevalence of frailty (25%) [18,19], our population was older and frailer. The impact on the mental and physical health status of preventive social distancing measures in frail community-dwelling older adults has been poorly described. Targeting such a vulnerable group is particularly relevant due to its higher risk of progressing to disability. Moreover, our study offers unique pre-post lockdown measures of PA.

Among the several public health challenges driven from the COVID-19 pandemic, promoting PA is particularly complex due to strict mobility restrictions, including access to public space (e.g., gyms, parks, civic centers, etc.), and social-distancing measures. These regulations precluded free and low-cost options to perform PA and might decrease motivation, hampering PA adherence. Interestingly, despite a long time since the last face-to-face visit, a remarkable proportion of our sample followed the personalized PA recommendations derived from the +ÀGIL Barcelona program. This reinforces the need of community-based programs to empower older adults for self-care [39].

The association between depressive symptoms and low PA levels has been previously described [40] and could be explained by generalized reduced activity, both in the cognitive/affective and behavioral realms. Depression negatively impacts lifestyle choices, and individuals with depressive symptoms tend to be less motivated, more sedentary and less physically fit than non-depressed ones [41,42]. In previous Spanish surveys during the lockdown, older persons showed less emotional distress and higher resilience to the pandemic than younger adults [43]. However, the profile of resilient individuals seemed to be characterized by more optimistic personality traits [44], a regular practice of vigorous and moderate PA, positive self-perceptions of ageing, less depressive symptoms [45,46] as well as less perceived loneliness during the lockdown [47]. It is possible that, in these surveys, vulnerable responders were not sufficiently represented. We also cannot exclude a bidirectional association between depressive symptoms and PA, because higher PA is associated with better physical and cognitive function [48], lower rates of frailty [49] and less depressive symptoms in community-dwelling older adults [50,51].

We also found a negative, independent association between fatigue and total PA levels and its improvement or maintenance during the lockdown. Fatigue, a subjective self-reported global tiredness and lack of energy [52], has been associated with lower physical and mental function, disability and mortality [53,54,55], and is one of the pillars of the frailty pre-disability concept [36,56]. Self-perceived fatigue can be a symptom of an underlying disease (e.g., cardiovascular, respiratory, psychiatric, etc.), but it has also been associated with an inactive lifestyle [57], so that a bidirectional causality, in the association between fatigue and PA, cannot be excluded, moreover because both fatigue and PA were collected in the same timeframe at the moment of the telephonic interview.

Social relationships are pivotal for healthy aging [58] and have been previously associated with a higher chance of maintaining physical health and longevity [59]. On the other hand, loneliness is a risk factor for physical and mental illness, fatigue and physical inactivity [60]. Consequently, social connections are essential to foster activity and PA in older adults and are an important component of PA group programs success [11,61]. Although during the first COVID-19 outbreak the population, especially older adults, may have progressively adapted to the new daily routines and limitations, this situation has a clear negative impact on social relationships and loneliness [62,63,64]. Tackling loneliness and social relationships requires specific implementation strategies [65], and these need to be adapted and implemented to promote the adherence to exercise programs [23,66], particularly in these challenging times. In summary, the complex interaction between depressive symptoms, physical function, social participation and activity deserves special attention in older adults [67], and this should be kept in mind for the post-lockdown and post-COVID-19 recovery plans.

Reading is a complex activity, which combines both cognitive and mental functions. Previous studies have reported that reading has a positive impact on stress, insomnia, depression symptoms and dementia development. Indeed, all of them related negatively with levels of physical activity [68]. Surprisingly, in our sample, although the group with preserved PA showed better physical function (either SPPB score or gait speed) and frailty (CFS), the association between frailty and PA was not significant in the multivariable models. Similarly, we found no association between previous cognitive impairment and PA. These negative findings might be attributable to the sample’s relative homogeneity enrolled in the +ÀGIL Barcelona program, where the frailty screening was an inclusion criterion [21].

### Limitations and Future Research Recommendations

We acknowledge the different limitations of our study. First, our pre-lockdown assessment cannot completely reflect the situation immediately pre-lockdown because of the time elapsed since the last face-to-face visit to the telephonic interview. However, it provides the added value of a longitudinal design. Second, the sample size was relatively small, although representing almost 50% of the whole +ÀGIL Barcelona sample. As for strengths, PA levels were assessed by the BPAAT scale, a short validated scale, with good psychometric properties, that was already part of the +ÀGIL Barcelona assessment, allowing us to track pre-post changes. In this same line, the study population had an extended pre-lockdown assessment, which offered a comprehensive sample characterization. The telephone interview was short, which is important for such a vulnerable population, who could get tired easily.

Considering that reduced PA is a key risk factor to increased frailty and disability, our results highlight the need to design and adapt strategies for community and home-based PA in older adults, particularly in challenging situations such as the ongoing COVID-19 pandemic. These strategies should likely take into account multifactorial contributors to reduced PA, such as mental status, social relationships and frailty-related symptoms such as fatigue. In light of COVID-19 pandemics, we also believe that there is an increased need for adapted digital solutions to provide PA in the community. These should be adapted through a broad system thinking strategy, particularly for vulnerable older adults with such multidomain contributors to inadequate PA.

## 5. Conclusions

In our sample, strict home lockdown due to the COVID-19 pandemic had determined a decrease in PA levels. Moreover, pre-lockdown mental health, frailty-related symptoms and social relationships were consistently associated with both PA levels during-lockdown and pre-post change. Our data could be used to design specific person-centered plans to maintain PA levels in frail older community-dwellers. However, larger population studies including dwelling older adults are needed to confirm our results.

## Figures and Tables

**Figure 1 ijerph-18-00808-f001:**
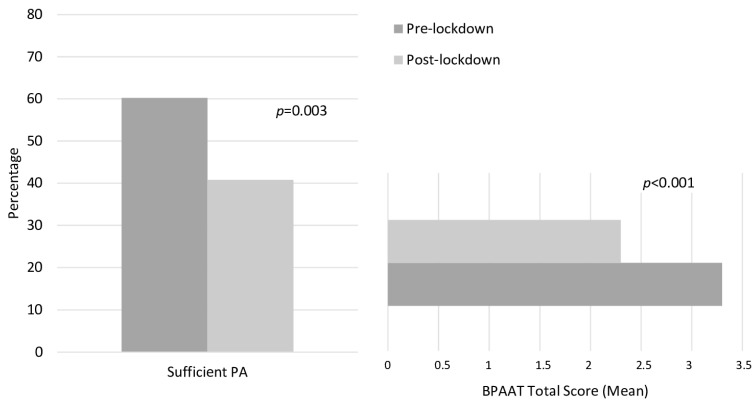
Effect of the strict lockdown due to COVID-19 pandemic on physical activity. The McNemar’s test for repeated samples was used for categorical variables. Paired sample *t*-test for repeated samples was used for continuous variables. Brief Physical Activity Assessment score ranged from 0–8 (≥4 points: sufficient active, 0–3: insufficient active).

**Table 1 ijerph-18-00808-t001:** Characteristics of the sample before the lockdown due to COVID-19.

Baseline Characteristics	Included *n* = 98	Reduction or Insufficient PA, *n* = 58 ^a^	Improvement or Sufficient PA, *n* = 40 ^a^	*p*-Value
Age, mean (SD)	82.4 (6.1)	82.2 (5.5)	82.8 (6.8)	0.606
Woman, % (*n*)	66.3 (65)	62.1 (36)	72.5 (29)	0.283
Lives alone, % (*n*)	54.1 (54)	56.9 (33)	50.0 (20)	0.501
Education, % (*n*)				
Illiterate Primary school Secondary school University degree	8.3 (8) 39.2 (38) 38.1 (37) 14.4 (14)	7.0 (4) 43.9 (25) 38.6 (22) 10.5 (6)	10.0 (4) 32.5 (13) 37.5 (15) 20.0 (8)	0.476
Falls in the last year, % (*n*)	28.6 (28)	27.6 (16)	30.0 (12)	0.795
Lawton Index ^b^, median (IQR)	5 (3–8)	4.5 (2–7)	7 (4–8)	0.012
Barthel Index ^c^, median (IQR)	95 (85–100)	92.5 (85–95)	95 (90–100)	0.091
Malnutrition risk ^d^, % (*n*)				
Normal nutrition status At risk of malnutrition Malnourished	79.0 (75) 19.0 (20) 1.1 (1)	73.2 (41) 25.0 (14) 1.8 (1)	87.2 (34) 12.8 (5) 0.0 (0)	0.227
Depressive symptoms ^e^, % (*n*)	21.9 (21)	30.4 (17)	10.0 (4)	0.017
Charlson Comorbidity Index, median (IQR)	2 (1–3)	2 (1–3)	1 (0–2)	0.041
Previous cognitive impairment or positive screening ^f^, % (*n*)	36.1 (35)	45.6 (26)	22.5 (9)	0.020
Number of drugs, mean (SD)	7.4 (3.5)	7.7 (3.4)	7.0 (3.5)	0.368
Clinical Frailty Scale—vulnerable or any degree of frailty, % (*n*)	63.3 (62)	67.2 (39)	57.5 (23)	0.326
SPPB ^g^, mean (SD)	8.3 (3.1)	7.9 (3.2)	8.8 (2.9)	0.202
Gait speed, median (IQR)	0.75 (0.58–0.92)	0.72 (0.66–0.77)	0.79 (0.72–0.86)	0.175
Sufficient physical activity, % (*n*)	60.2 (59)	51.7 (30)	72.5 (29)	0.039

PA: Physical Activity. IQR: interquartile range, SD: standard deviation. Student’s *t*-test or the Mann-Whitney U-test were used for continuous variables as appropriate and Chi-square test for categorical. ^a^ Change in PA level: Improve an insufficient or maintain a sufficient PA level vs. reduction or maintain insufficient PA level. Brief PA Assessment score, range from 0–8 (≥4 points: sufficient active, 0–3: insufficient active). ^b^ Independence for activities of daily living, Barthel index: range from 0–100. ^c^ Independence for instrumental activities of daily living, Lawton index: range from 0–8. ^d^ Mini-Nutritional Assessment Short form score: range from 0–14 points (0–7: Malnourished, 8–11: At risk of malnutrition, 12–14: Normal). ^e^ Geriatric Depression Scale Yesavage: range from 0–15 points (>5 points: depression). ^f^ Previous diagnosis of cognitive impairment or dementia or positive screening performed with Minicog©. Minicog© range 0–5 (<3 positive screening for cognitive impairment). ^g^ Short Physical Performance Battery, range from 0–12 (<10 points: frailty).

**Table 2 ijerph-18-00808-t002:** Description of characteristics of the sample during the lockdown due to COVID-19.

Baseline Characteristics	Included *n* = 98	Reduction or Insufficient PA, *n* = 58 ^a^	Improvement or Sufficient PA, *n* = 40 ^a^	*p*-Value
Lives alone, % (*n*)	38.1 (37)	38.6 (22)	37.5 (15)	0.913
Maintained daily social contact (any type), % (*n*)	79.6 (78)	75.9 (44)	85.0 (34)	0.270
Social contact different than family, % (*n*)	46.9 (46)	37.9 (22)	60.0 (24)	0.031
Any new health concerns ^b^, % (*n*)	39.8 (39)	48.3 (28)	27.5 (11)	0.039
Sought medical attention, % (*n*)	56.4 (22)	57.1 (16)	54.6 (6)	0.883
Following +ÀGIL PA recommendation, % (*n*)	22.5 (22)	8.6 (5)	42.5 (17)	<0.001
Obstacles to desired PA, % (*n*)				
Apathy Falls Fatigue Functional impairment Lockdown situation Medical condition incident No time	30.3 (10) 3.0 (1) 15.2 (5) 3.0 (1) 15.2 (5) 21.2 (7) 12.2 (4)	29.6 (8) 3.7 (1) 18.5 (5) 3.7 (1) 11.0 (3) 25.9 (7) 7.4 (2)	33.3 (2) 0.0 (0) 0.0 (0) 0.0 (0) 33.3 (2) 0.0 (0) 33.3 (2)	0.362
Vigorous PA (one-twice/wk) ^c^	5.2 (5)	3.5 (2)	7.7 (3)	0.354
Moderate PA ^d^, % (*n*)				
≥5 times/wk 3–4 times/wk 1–2 times/wk Never	39.8 (39) 19.4 (19) 26.5 (26) 14.3 (14)	0.0 (0) 31.0 (18) 44.8 (26) 24.1 (14)	97.5 (39) 2.5 (1) 0.0 (0) 0.0 (0)	<0.001
Self-reported fatigue	38.1 (37)	49.1 (28)	22.5 (9)	0.008
Activities to stay active during the lockdown ^e^, % (*n*)				
Housework Leisure activities ^f^ Music/TV Provide care Reading Social contact Use of technology	45.9 (45) 36.7 (36) 69.4 (68) 5.1 (5) 26.5 (26) 10.2(10) 5.1 (5)	37.9 (22) 32.8 (19) 74.1 (43) 8.6 (5) 17.2 (10) 8.6 (5) 1.7 (1)	57.5 (23) 42.5 (17) 62.5 (25) 0.0 (0) 40.0 (16) 12.5 (5) 10.0 (4)	0.056 0.326 0.219 0.057 0.012 0.533 0.067

PA: Physical activity. Wk: week. Chi-square test was performed to analyze the difference between categorical variables. ^a^ Change in PA level: Improve insufficient or maintain sufficient PA level vs reduction or maintain insufficient PA level. Brief Physical Activity Assessment score, range from 0–8 (≥4 points: sufficient active, 0–3: insufficient active). ^b^ Acute health concern: diarrhea, urinary infection, allergies. ^c^ ≥20 min or more of jogging (mainly in the house), heavy lifting, etc. ^d^ ≥30 min walking that increases heart rate or breath harder than normal. ^e^ Not mutually exclusive. ^f^ Painting, crafts, table games, urban gardening.

**Table 3 ijerph-18-00808-t003:** Association of pre-lockdown characteristics and total physical activity ^a^ during the lockdown or pre-post improvement or maintenance of sufficient physical activity ^a^.

Prelockdown Characteristics (from the Last Available Assessment)	Linear Regression			Logistic Regression		
	BPAAT during-Lockdown Total Score ^a^			Improve or Maintain Sufficient PA during vs. Prelockdown ^b^		
	B	(95% CI)	*p*	OR	(95% CI)	*p*
Age	0.01	−0.04; 0.05	0.752	1.03	0.95; 1.12	0.494
Female	0.30	−0.31; 0.91	0.336	2.61	0.85; 8.04	0.094
Education	0.03	−0.13, 0.18	0.705	0.87	0.65; 1.17	0.370
Depressive symptoms (pre-lockdown) ^c^	−1.15	−1.89; −0.41	0.003	0.12	0.02; 0.55	0.006
Social contact different than family (during the lockdown)	0.99	0.41; 1.57	0.001	5.07	1.60; 16.08	0.006
Self-reported fatigue	−1.25	−1.87; −0.63	<0.001	0.11	0.03; 0.44	0.002
Reading to stay active (during the lockdown)	0.74	0.08; 1.39	0.028	6.29	1.66; 23.90	0.007
Lives alone (pre-lockdown)	−1.30	−2.14; −0.46	0.003	-		-
Lives alone (during the lockdown)	−0.78	−1.74; 0.07	0.073	-	-	-
Diagnosis of cognitive impairment (pre-lockdown) ^d^	-	-	-	0.29	0.08; 1.06	0.061

Stepwise multivariable linear and logistic regression were performed as appropriate. Age, sex and education level were set as lockterm in both cases. Variables with empty cells were not included in the final model. PA: Physical Activity. ^a^ Brief Physical Activity Assessment score, range from 0–8. ^b^ Change in physical activity level: Improve insufficient or maintain sufficient physical activity level vs reduction or maintain insufficient physical activity level. ^c^ Geriatric Depression Scale Yesavage: range from 0–15 points (>5 points: depression). ^d^ Previous diagnosis of cognitive impairment or dementia or last Minicog© assessment <3. Minicog© range 0–5 (<3 positive screening for cognitive impairment).

## Data Availability

The data presented in this study are available on request from the corresponding author. The data are not publicly available due to ethical reasons.

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
