# Peer review of "Depressive Symptoms, Fatigue and Social Relationships Influenced Physical Activity in Frail Older Community-Dwellers during the Spanish Lockdown due to the COVID-19 Pandemic"

_ijerph, 2021, doi:10.3390/ijerph18020808_

Round 1
Reviewer 1 Report
I must thank the authors for their work and effort. However, I suggest a series of changes and deepening.
1.- Introduction: It is too short. The authors deal with different concepts and must at least introduce a brief description of each of them, as well as describe the background that is related to what has been said.
2.- Method: This section is well resolved.
3.- Results: The results are well executed.
4.- Discussion: It is well executed but unbalanced in relation to the introduction. When the introduction is modified by adding background information, the discussion will also be modified. This should be taken into account by the authors.
5.- Conclusion: The authors should add limitations and foresight in this section. In addition, they should detail what the theoretical and practical implications of the study are.
Author Response
Manuscript ID: ijerph-1065856
Title: Depressive symptoms, fatigue and social relationships influenced physical activity in frail older community-dwellers during the Spanish lockdown due to the COVID-19 pandemic
We would like to thank the reviewers for their valuable feedback, which has help us to improve the quality of the manuscript. We have provided a point-by-point response.
Reviewer # 1
1.- Introduction: It is too short. The authors deal with different concepts and must at least introduce a brief description of each of them, as well as describe the background that is related to what has been said.
Answer: We thank the reviewer to point out this aspect. We now added a paragraph highlighting the importance of taking into account variables from the comprehensive geriatric approach as a potential predictor of complex, multi-factorial problems of older adults: in this line, we remark that the interaction of different domains could also be a basis for changes in physical activity. We also state the potential relevance of this work on predictors of change in physical activity to possibly design further strategies to resume activity in older adults.
2.- Method: This section is well resolved.
Answer: Thank you for the positive feedback.
3.- Results: The results are well executed.
Answer: We thank the reviewer for the positive feedback.
4.- Discussion: It is well executed but unbalanced in relation to the introduction. When the introduction is modified by adding background information, the discussion will also be modified. This should be taken into account by the authors.
Answer: Thank you for raising this comment, information in both sections has been revised and adjusted.
5.- Conclusion: The authors should add limitations and foresight in this section. In addition, they should detail what the theoretical and practical implications of the study are.
Answer: The limitations and strengths of the present work just precede the Conclusions, where, according to your comment, we now inform about the potential relevance of our data to design specific person-centered plans to maintain PA levels in frail older community-dwellers, and the need of larger studies including dwelling older adults to reinforce our results.
Reviewer 2 Report
General Comments
The purpose of this paper was to examine how the COVID lockdown affected PA in older adults from Spain, and what factors predicted maintenance or improvement of PA. The paper is timely and relevant, and it is part of an important study on aging out of Spain. I enjoyed reading it and it should make a nice contribution to the literature. To make this paper publishable, additional editing is needed, so I recommend sending it to an editor prior to the next submission. I have tried to help with some suggestions in the “Specific Comments” section below.
Specific Comments
ABSTRACT
Line 27: change “to identify” to “identified”
Lines 29-30: Not sure what is meant by “if a previous 12 months comprehensive geriatric assessment was available. Please clarify.
Line 33: edit to read: “..participants (32.2%) were not meeting sufficient PA levels…
Line 35: edit to read: “…whereas maintaining social networks…”
Line 39: change “revert” to “maintain”
INTRO
Line 43: edit to: “..pandemic has had a dramatic impact on the population…”
Line 45: change “growingly” to “growing”
Line 46: change “In” to “During”
Line 47, revise to: “..homes except to attend work, essential…shop for food and take care of…”
Line 55: remove “have reason to “
Line 58, edit to: “…these have caused a radical and sudden change in people’s lifestyles, in particular to PA levels.”
Line 64, edit: “…new challenges to the approach…”
Line 65, edit: “…impact and consequences of decreased daily activities and social contacts…”
Line 67, edit: “..relevant due to the increased risk of…”
METHODS
Study Population
Lines 76-77, edit: “…In brief, the program enrolls non-disabled frail older adults, based on…”
Line 79: Remove “S” from weeks (should read week).
Procedure & Data Collection
Line 90, edit: “..a follow-up visit via phone.. with each participant in the..”
Line 91, edit: “…during the 12 months prior to the lockdown…”
Line 92, edit: “..was not able to complete the phone call, a follow-up…”
Measure of PA
Line 98, edit: “..Level of PA..” (remove “The” to start sentence)
Line 100, edit: “…allowing the ability to distinguish…”
Lines 105-105: remove “The” from the start of the sentence; start with “Total” and “Improvement”
What is the outcome measure of the PA variable (e.g., minutes of PA, meets/does not meet recommendations? Add more information about this measure.
Covariates
Need to add reliability and validity information for each of the scales used in this study in this section AND need to describe what scores “mean” and what high or low means for each of the scales (right now, only the range of scores is provided—no information about high or low anchor is provided).
RESULTS
Line 145: change “accepted” to “agreed”
Line 160—put the figure caption on the same page as the figure if possible
Line 162: change “range” to “ranged”
Lines 170-171: Clarify the following statement: “..and to perform other leisure activities, such as reading, as a strategy to stay active.” It doesn’t seem to me that reading will enable someone to stay physically active—maybe mentally active?!?
Lines 209-214: Reword this section to ensure results are clearly described
Lines 213-214: PA was associated with reading? I’m guessing you addressed some reasons for this in the discussion?
Tables 1 & 2: Put the letters indicating significance NEXT TO THE P-Value, rather than by the measure
DISCUSSION
Line 219: change “lower” to “less”
Line 221: Finish the sentence: “…fatigue had an inverse association with…”
Line 225: replace “was driven from” to “used”
Line 229: change “frailer” to “more frail”
Line 229-230: Clarify what you mean by the following sentence: “…the impact of preventive social distancing measures in frail… has been poorly described.”
Line 236: remove “the” (use: hampering adherence to PA)
Line 238: replace “remarks” with “reinforces”
Lines 242-243, edit: “…previously described [39] and could be explained by generalized reduced activity
Line 253: You mention “absolute and relative” PA levels for the first time here. What is meant by that? Define it here or earlier in the paper when you explain PA calculations.
Lines 264-265: re-word and clarify what you mean in this sentence
Line 267, remove “also in”
Lines 276-284: Add a subheading to denote limitations within the discussion section; add some recommendations for future research to this section as well.
Author Response
Manuscript ID: ijerph-1065856
Title: Depressive symptoms, fatigue and social relationships influenced physical activity in frail older community-dwellers during the Spanish lockdown due to the COVID-19 pandemic
We would like to thank the reviewers for their valuable feedback, which has help us to improve the quality of the manuscript. We have provided a point-by-point response.
Reviewer # 2
- Line 27: change “to identify” to “identified”.
Answer: Thank you for the suggestion. Specific change has been introduced (line 27).
- Lines 29-30: Not sure what is meant by “if a previous 12 months’ comprehensive geriatric assessment was available. Please clarify.
Answer: We agree with the reviewer this must be confusing. Due to the limit of space in the abstract and as it is well explained in the Methods section. We have thus decided to remove from the Abstract the phrase “… if a previous 12-months comprehensive geriatric assessment was available”.
To clarify it means that we performed a follow-up visit via phone to each participant included in the +ÀGIL Barcelona program who had been assessed face-to-face during the 12 months before prior to the lockdown (either as the baseline, 3- or 6-month visit).
- Line 33: edit to read: “..participants (32.2%) were not meeting sufficient PA levels…
Answer: Thank you for the suggestion. Specific change has been introduced (line 33).
- Line 35: edit to read: “…whereas maintaining social networks…”
Answer: Thank you for the suggestion. Specific change has been introduced (line 35).
- Line 39: change “revert” to “maintain”
Answer: Thank you for the suggestion. Specific change has been introduced (line 38).
- Line 43: edit to: “..pandemic has had a dramatic impact on the population…”
Answer: Thank you for the suggestion. Specific change has been introduced (line 43).
- Line 45: change “growingly” to “growing”
Answer: Thank you for the suggestion. This sentence has been omitted.
- Line 46: change “In” to “During”
Answer: Thank you for the suggestion. Specific change has been introduced (line 46).
- Line 47, revise to: “..homes except to attend work, essential…shop for food and take care of…”
Answer: Thank you for the suggestion. Specific change has been introduced (line 47).
- Line 55: remove “have reason to “
Answer: Thank you for the suggestion. Specific change has been done.
- Line 58, edit to: “…these have caused a radical and sudden change in people’s lifestyles, in particular to PA levels.”
Answer: Thank you for the suggestion. Specific change has been introduced (line 76).
- Line 64, edit: “…new challenges to the approach…”
Answer: Thank you for the suggestion. We have added more information to the text (line 83) and now it reads as follow: “.. new challenges to community-dwelling frail older adults' approach and care.”
- Line 65, edit: “…impact and consequences of decreased daily activities and social contacts…”
Answer: Thank you for the suggestion. This change has been introduced (line 84).
- Line 67, edit: “..relevant due to the increased risk of…”
Answer: Thank you for the suggestion. This change has been introduced (line 86).
- Lines 76-77, edit: “…In brief, the program enrolls non-disabled frail older adults, based on…”
Answer: Thank you for the suggestion. This change has been introduced (line 94).
- Line 79: Remove “S” from weeks (should read week).
Answer: Thank you for the suggestion. This change has been introduced (line 96).
- Line 90, edit: “..a follow-up visit via phone.. with each participant in the..”
Answer: Thank you for the suggestion. This change has been introduced (line 104).
- Line 91, edit: “…during the 12 months prior to the lockdown…”
Answer: Thank you for the suggestion. This change has been introduced (line 105).
- Line 92, edit: “..was not able to complete the phone call, a follow-up…”
Answer: Thank you for the suggestion. We have reworded this sentence and now it reads as follow: “In case the participant could not complete the phone call assessment, a self-identified proxy or caregiver answered the follow-up interview.”
- Line 98, edit: “..Level of PA..” (remove “The” to start sentence)
Answer: Thank you for the suggestion. This change has been introduced (line 110).
- Line 100, edit: “…allowing the ability to distinguish…”
Answer: Thank you for the suggestion. This change has been introduced (line 114).
- Lines 105-105: remove “The” from the start of the sentence; start with “Total” and “Improvement”
Answer: Thank you for the suggestion. This change has been introduced (lines 117-118).
- What is the outcome measure of the PA variable (e.g., minutes of PA, meets/does not meet recommendations? Add more information about this measure.
Answer: We agree with the reviewer that this issue deserves further explanation. We have therefore added more information to the text and now it reads as follow:
“The BPAAT is a two-question tool. The first item explores the frequency and duration of PA at vigorous intensity, and the second item assesses the frequency and PA duration at moderate-intensity during a typical week. The BPPAT scoring algorithm was designed to identify whether patients meet or not PA recommendations through the combination of both questions. Its total score ranges from 0 to 8, allowing the ability to distinguish "sufficiently active" (20 minutes of vigorous-intensity ≥3 times/week or 30 minutes of moderate-intensity ≥5 times/week or ≥5 times/week of any combination of moderate or vigorous PA, scores 4-8 points) from "insufficiently active" participants (do not meet any previous recommendation, scores 0-3 points). Previous studies report a reliability 0.76 validity of 0.71 [24]”
- Need to add reliability and validity information for each of the scales used in this study in this section AND need to describe what scores “mean” and what high or low means for each of the scales (right now, only the range of scores is provided—no information about high or low anchor is provided).
Answer: We agree with the reviewer that this issue deserves further explanation. We have therefore added more information of each scale to the text. The scales used in the face-to-face assessment have been widely used as part of the Comprehensive Geriatric Assessment, including functional, physical, cognitive, nutritional, affective, and social status. All the scales and measures are, of course, validated, and the specific references to the validation work is provided now in the manuscript, in the sections: “Measure of Physical activity” and “Covariates”. Given the relevance of PA measure as the outcome of this analysis, we provided detailed numeric information about reliability and validity of this scale in the corresponding section.
- Line 145: change “accepted” to “agreed”
Answer: Thank you for the suggestion. This change has been performed.
- Line 160—put the figure caption on the same page as the figure if possible
Answer: Thank you for the suggestion. This change has been performed (line 354).
- Line 162: change “range” to “ranged”
Answer: Thank you for the suggestion. This change has been performed.
- Lines 170-171: Clarify the following statement: “..and to perform other leisure activities, such as reading, as a strategy to stay active.” It doesn’t seem to me that reading will enable someone to stay physically active—maybe mentally active?!?
Answer: Thank you for the suggestion. We added to the text (line 390): “to stay physically or mentally active”.
- Lines 209-214: Reword this section to ensure results are clearly described
Answer: We agree with the reviewer that this issue deserves further explanation. We have therefore added more information in the text:
“In multivariable models, living alone (before the lockdown (ß= -1.30, 95%CI -2.14 - -0.46, p=0.003)), previous depressive symptoms (ß= -1.15, 95%CI -1.89 - -0.41, p=0.003) and self-reported fatigue during the COVID-19 outbreak (ß= -1.25, 95%CI -1.87 - -0.63, p<0.001) were inversely associated with PA levels (BPAAT total score) during the lockdown. Having social contact with people different from family (ß=0.99, 95%CI 0.41-1.57, p=0.001) and performing reading activities during the lockdown (ß=0.74, 95%CI 0.08-1.39, p=0.028) were associated with higher BPAAT scores during the lockdown (Table 3)”.
- Lines 213-214: PA was associated with reading? I’m guessing you addressed some reasons for this in the discussion?
Answer: Thank you for the suggestion. We acknowledge this point deserved further explanation. We added to the manuscript the next paragraph (line 498):
“Reading is a complex activity, which combines both cognitive and metal functions. Previous studies have reported that reading has a positive impact on stress, insomnia, depression symptoms and dementia development. Indeed, all of them related negatively with levels of physical activity.”
- Tables 1 & 2: Put the letters indicating significance NEXT TO THEP-Value, rather than by the measure
Answer: Thank you for the suggestion. The letter next to the measure is a resume explanation rather than the p-value. The significant p-values are highlighted in bold.
- Line 219: change “lower” to “less”
Answer: Thank you for the suggestion. This change has been performed (line 444).
- Line 221: Finish the sentence: “…fatigue had an inverse association with…”
Answer: Thank you for the suggestion. This change has been performed (line 446). We finished the sentence as “ … with PA levels.”
- Line 225: replace “was driven from” to “used”
Answer: Thank you for the suggestion. This change has been performed (line 252).
- Line 229: change “frailer” to “more frail”
Answer: Thank you for the suggestion. This change has been performed (line 449).
- Line 229-230: Clarify what you mean by the following sentence: “…the impact of preventive social distancing measures in frail… has been poorly described.”
Answer: Thank you for the suggestion. We have reworded the phrase as follow (line 452). “So, that the impact on the mental and physical health status of preventive social distancing measures in frail community-dwelling older adults has been poorly described.”
- Line 236: remove “the” (use: hampering adherence to PA)
Answer: Thank you for the suggestion. This change has been performed (line 457).
- Line 238: replace “remarks” with “reinforces”
Answer: Thank you for the suggestion. This change has been performed (line 459).
- Lines 242-243, edit: “…previously described [39] and could be explained by generalized reduced activity
Answer: Thank you for the suggestion. This change has been performed (line 474).
- Line 253: You mention “absolute and relative” PA levels for the first time here. What is meant by that? Define it here or earlier in the paper when you explain PA calculations.
Answer: Thank you for the suggestion. We agree with the reviewer this phrase need to be reword. We changed in the text as follow (line 483):
“We also found a negative, independent, association between fatigue and total absolute and relative PA levels and its it improvement or maintenance during the lockdown.”
- Lines 264-265: re-word and clarify what you mean in this sentence
Answer: Thank you for the suggestion. We had reworded the sentence as following (line 492):
“Although during the first COVID-19 outbreak the population, especially older adults, may have progressively adapted to the new daily routines and limitations, this situation has is having a clear negative impact on social relationships and loneliness.”
- Line 267, remove “also in”
Answer: Thank you for the suggestion, change was already introducing (line 486).
- Lines 276-284: Add a subheading to denote limitations within the discussion section; add some recommendations for future research to this section as well.
Answer: Thank you for the suggestion. We added the subheading: Limitations and future research recommendations; as well as some insights in this section.
Reviewer 3 Report
This research describes how the lockdown modified older adults' physical activity (PA) and evaluates the effect of lockdown on PA levels and to identify predictors of sufficient/insufficient PA in frail older community-dwellers who participated in a pre-lockdown intervention program.
Comments
The present work presents two very valuable qualities.
The first is to have pre- and post-lockdown measures.
This offers a very interesting overview of how the activities associated with the intervention program have evolved and its correlations with the rest of health and functionality indicators.
The second is the containment with which the authors interpret the results. The limitations of the study prevent making far-reaching inferences. The authors maintain their interpretations in a adequate level.
The main limitation of their work also comes from the design itself. The results of this study does not allow to propose concrete strategies to increase the presence of physical activity predictors, only to point out that certain deficiencies (social relations, leisure activities such as reading or absence of subjective symptoms of health and energy) are related to the reduction or insufficient practice of physical activity and functionality. Nevertheless, this does not detract from the work whose objectives are well specified and conform to the statistical analyses carried out.
Author Response
Manuscript ID: ijerph-1065856
Title: Depressive symptoms, fatigue and social relationships influenced physical activity in frail older community-dwellers during the Spanish lockdown due to the COVID-19 pandemic
We would like to thank the reviewers for their valuable feedback, which has help us to improve the quality of the manuscript. We have provided a point-by-point response.
Reviewer # 3.
The present work presents two very valuable qualities. The first is to have pre- and post-lockdown measures. This offers a very interesting overview of how the activities associated with the intervention program have evolved and its correlations with the rest of health and functionality indicators. The second is the containment with which the authors interpret the results. The limitations of the study prevent making far-reaching inferences. The authors maintain their interpretations in an adequate level.
The main limitation of their work also comes from the design itself. The results of this study does not allow to propose concrete strategies to increase the presence of physical activity predictors, only to point out that certain deficiencies (social relations, leisure activities such as reading or absence of subjective symptoms of health and energy) are related to the reduction or insufficient practice of physical activity and functionality. Nevertheless, this does not detract from the work whose objectives are well specified and conform to the statistical analyses carried out.
Answer: Thank you very much for your positive feedback. We appreciate your comments. We agree that bidirectional causality in the association between physical activity and certain deficits might exit. It has been discussed in the manuscript and also, future directions in research and conclusions have been improved based on your insights.
Round 2
Reviewer 1 Report
Dear Authors. Thank you for making the requested changes. The writing is more proportionate and better explained.
Congratulations.